# On 3D Reconstruction Pre-training to Improve Adversarial Robustness

## Abstract

Ensuring robustness of image classifiers against worst-case adversarial perturbations has been challenging. A promising idea is to build models that use robust features, instead of non-robust features that nevertheless generalize to test sets. One of the most effective methods for inducing such robust features is a type of data augmentation that uses adversarial examples during training. Here, inspired by studies of human vision, we explore a synthesis of this approach by leveraging a causal property underlying image formation: the 3D geometry of objects and how it projects to images. We combine adversarial training with a weight initialization that encodes prior knowledge about 3D objects, which is achieved via 3D reconstruction pre-training. We evaluate our approach using two different datasets and compare it to alternative non-3D pre-training protocols. To systematically explore the effect of 3D pre-training, we introduce a novel dataset called Geon3D, which consists of simple shapes that nevertheless capture variation in multiple distinct dimensions of geometry. We find that while 3D reconstruction pre-training does not improve robustness for the simplest dataset setting we consider (Geon3D on a clean background), it improves upon adversarial training in more realistic (Geon3D with textured background) and challenging (spurious correlations between shape and background textures) dataset conditions, as well as on a dataset with a more complex distribution of shapes (ShapeNet). Furthermore, we show that the benefit of using 3D-based pre-training outperforms 2D-based pre-training on ShapeNet. We hope that these results encourage further investigation of the benefits of 3D vision for adversarial robustness.

## 1 Introduction

Adversarial examples were first reported about a decade ago Szegedy et al. (2014). Despite tremendous research efforts since then, adversarial robustness remains perhaps the most important challenge to safe, real-world deployment of modern computer vision systems. Many proposals to defend against adversarial perturbations are later found to be broken Athalye et al. (2018). A promising defense method that has withstood scrutiny is adversarial training Madry et al. (2018). Recent work Ilyas et al. (2019) suggests that adversarial training leads to machine vision systems that learn robust, instead of "non-robust" features that nevertheless generalize to held-out test sets.

Yet, existing work has so far did not attempt to directly or independently design or leverage robust features in the context of adversarial robustness. Prior work extends adversarial training via surrogate-loss Zhang et al. (2019), using additional unlabelled data Carmon et al. (2019); Alayrac et al. (2019), or pre-training on more natural images Hendrycks et al. (2019).

In this work, we turn to recent advances in 3D computer vision (especially implicit neural representations Mescheder et al. (2019); Park et al. (2019)) to test the following hypothesis: If adversarial training leads models to learn robust features in datasets, then we should be able to improve its performance by leveraging such robust features more directly. In this study, we consider one class of such robust features, the 3D scene parameters (object shape and pose) that causally underlie image formation: Can we improve adversarial training with a weight initialization based on a 3D reconstruction pre-training that encodes such prior

knowledge of 3D object geometry and image formation? To do so, we consider recent 3D reconstruction models that are equipped with an image encoder based on Convolutional Neural Networks (CNNs). The goal of such an image encoder is to produce useful representations for 3D reconstruction, and therefore is expected to encode the representation of 3D objects, which can be leveraged as a better weight initialization for adversarial training, as summarized in Figure 1.

Our proposal is also inspired by a basic observation about how human vision works: Unlike much of the adversarial robustness research in machine vision that focuses on the task of classification, the human visual system also recovers rich, 3D geometry, including object shape and pose, from single image. This ability to make inferences about the underlying scene structure from input images—also known as analysis-by-synthesis—is thought to be critical for the robustness of biological vision (Yuille & Kersten, 2006; Mumford, 1994).

Standard benchmark datasets for adversarial robustness research include MNIST, CIFAR-10, and Tiny-ImageNet. However, these datasets are not suitable to address our question, as they are not designed to be used for 3D reconstruction tasks. To understand the interplay between shape datasets used for 3D-based pre-training and the performance of adversarial training, we introduce *Geon3D*—a novel dataset comprised of simple yet realistic shape variations, derived from the human object recognition hypothesis called *geon theory* (Biederman, 1987).

Using Geon3D as a bridge from simple objects to more complex real shape objects like ShapeNet, we systematically perform experiments varying the complexity of the shape dataset. We first find that 3D-based pre-training does not improve the performance of adversarial training in the simplest shape dataset we consider (Geon3D with black background). However, when in a more realistic variation of Geon3D with textured backgrounds, we find 3D-based pre-training strengthens $L_\infty$-based adversarial training. When we introduce spurious correlation between shape and background, 3D-based pre-training outperforms vanilla adversarial training for both $L_\infty$ and $L_2$ threat models. We further confirm that this trend continues to hold for more complicated shape objects, namely ShapeNet dataset Chang et al. (2015). Crucially, we show that the benefit of 3D-based pre-training outperforms 2D-based pre-training on ShapeNet. While our study is limited to shape datasets, as 3D reconstruction techniques improve to deal with increasingly more realistic, complicated settings, we hope our study serves as a first step towards better understanding the relationship between 3D vision and adversarial robustness.

## 2 3D reconstruction as pre-training

Recently, there has been significant progress in learning-based approaches to 3D reconstruction, where the data representation can be classified into voxels (Choy et al., 2016; Riegler et al., 2017), point clouds (Fan et al., 2017; Achlioptas et al., 2018), mesh (Kato et al., 2018; Groueix et al., 2018), and neural implicit representations (Mescheder et al., 2019; Chen & Zhang, 2019; Park et al., 2019; Sitzmann et al., 2019). In this paper, we are interested in methods that can be used to pre-train an image encoder so that we can use the weights of the pre-trained image encoder as initialization for adversarial training of image classifiers. For this purpose, we avoid 3D reconstruction models based on voxels, point clouds, and 3D meshes, since they are not easily transferable to image classification settings. Luckily, neural implicit representation allows the community to develop a class of models that only requires 2D supervision.

Specifically, we use two recent 3D reconstruction models: Differentiable Volumetric Rendering (**DVR**) (Niemeyer et al., 2020) and **pixelNeRF** Yu et al. (2021), both of which consist of a CNN-based image encoder and a differentiable neural rendering module. While implicit representation of 3D objects are done by a neural network-based rendering module in the 3D reconstruction model, we hypothesize that an image encoder of the 3D reconstruction model is biased towards producing an encoded representation that is useful for 3D geometry understanding. The main object of our study is to see to what extent we can leverage 3D reconstruction pre-training to improve adversarial robustness.

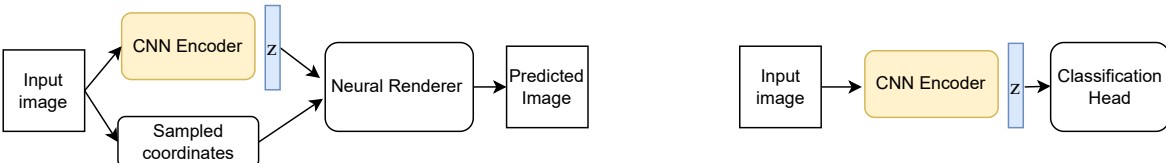

(a) 2D supervision-based 3D reconstruction pipeline    (b) Adversarial training after 3D reconstruction pre-training

Figure 1: (a) A class of 3D reconstruction models we are interested in is presented, where a CNN encoder is used to condition the 3D reconstruction model on shape features of 2D input images. (b) To leverage 3D-based pre-training, we extract the weights from the CNN encoder that is pre-trained on 3D reconstruction and use them as initialization for adversarial training on 2D rendered images of 3D objects. The goal of this paper is to investigate the effect of 3D reconstruction pre-training of these image encoders on adversarial robustness.

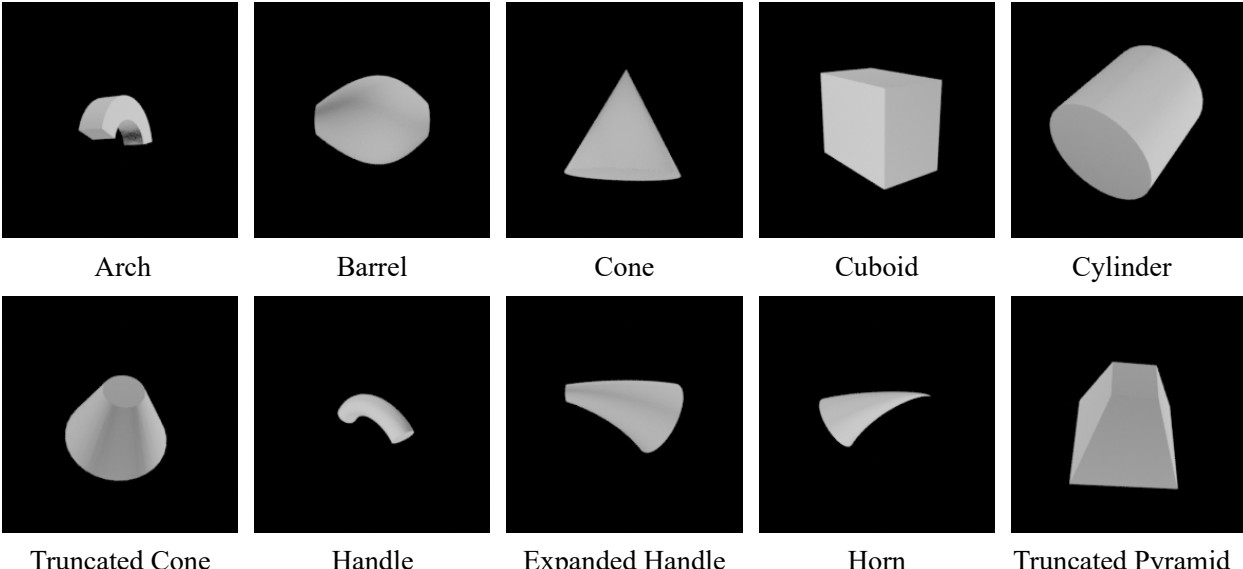

| Arch | Barrel | Cone | Cuboid | Cylinder |
| Truncated Cone | Handle | Expanded Handle | Horn | Truncated Pyramid |

Figure 2: Examples of 10 Geon categories from Geon3D. The full list of 40 Geons we construct (Geon3D-40) is provided in the Appendix.

## 2.1 Problem setup for 3D reconstruction

Both DVR and pixelNeRF are based on neural implicit representations. DVR learns the occupancy field via neural network, and represents objects via the zero-level set, which is found via ray-marching. The points corresponding to the zero-level are used to query a texture network, which produces RGB values as rendered images. The image encoder of DVR is used to condition the occupancy network and texture network. PixelNeRF is based on NeRF, which learns radiance field via neural network. Given a spatial point and viewing direction, the radiance field returns the density and RGB color. PixelNeRF additionally conditions NeRF by the local image features produced by the image encoder. The radiance field can then be rendered by volumetric rendering. We note that only DVR requires object masks, and pixelNeRF can be trained fully based on 2D images and camera matrices. For more details on the problem setup and training, we refer the readers to the supplementary materials.

## 3 Geon3D Benchmark

The concept of *geons*—or *geometric ions*—was originally introduced by Biederman as the building block for his Recognition-by-Components (RBC) Theory (Biederman, 1987). The RBC theory argues that human shape perception segments an object at regions of sharp concavity, modeling an object as a composition of geons—a subset of generalized cylinders (Binford, 1971). Similar to generalized cylinders, each geon is defined by its axis function, cross-section shape, and sweep function. To reduce the possible set of generalized cylinders, Biederman considered the properties of the human visual system. He noted that the human visual system is better at distinguishing between straight and curved lines than at estimating curvature; detecting parallelism than estimating the angle between lines; and distinguishing between vertex types such as an arrow, Y, and L-junction (Ikeuchi, 2014).

Table 1: Latent features of Geons. S: Straight, C: Curved, Co: Constant, M: Monotonic, EC: Expand and Contract, CE: Contract and Expand, T: Truncated, P: End in a point, CS: End as a curved surface

| Feature | Values |
|---|---|
| Axis | S, C |
| Cross-section | S, C |
| Sweep function | Co, M, EC, CE |
| Termination | T, P, CS |

Table 2: Similar Geon categories, where only a single feature differs out of four shape features. "T." stands for "Truncated". "E." stands for "Expanded".

| Geon Category | Difference |
|---|---|
| Cone vs. Horn | Axis |
| Handle vs. Arch | Cross-section |
| Cuboid vs. Cyllinder | Cross-section |
| T. Pyramid vs. T. Cone | Cross-section |
| Cuboid vs. Pyramid | Sweep function |
| Barrel vs. T. Cone | Sweep function |
| Horn vs. E. Handle | Termination |

The paper is not focused on the validity of the RBC theory. Instead, we wish to build upon the way Biederman characterized these geons. Biederman proposed using two to four values to characterize each feature of the geons. Namely, the axis can be straight or curved; the shape of cross section can be straight-edged or curved-edged; the sweep function can be constant, monotonically increasing / decreasing, monotonically increasing and then decreasing (i.e. expand and contract), or monotonically decreasing and then increasing (i.e. contract and expand); the termination can be truncated, end in a point, or end as a curved surface. A summary of these dimensions is given in Table 1.

Representative geon classes are shown in Figure 2. For example, the "Arch" class is uniquely characterized by its curved axis, straight-edged cross section, constant sweep function, and truncated termination. These values of geon features are *nonaccidental*—we can determine whether the axis is straight or curved from almost any viewpoint, except for a few *accidental* cases. For instance, an arch-like curve in the 3D space is perceived as a straight line only when the viewpoint is aligned in a way that the curvature vanishes.

**Data Preparation**

We construct each Geon using Blender —an open-source 3D computer graphics software Blender (2021).

An advantage of Geons over other geometric primitives such as superquadrics Barr (1981) is that the shape categorization of Geons is qualitative rather than quantitative. Thus, each Geon category affords a high degree of in-class shape deformation, as long as the four defining features of each shape class remains the same. Such flexibility allows us to construct a number of different 3D model instances for each Geon class by expanding or shrinking the object along the x, y, or z-axis. In our experiments, for each axis, we evenly sample the 11 scaling parameters from the interval [0.5, ..., 1.5] with a step size 0.1, resulting in 1331 model instances for each Geon category.

**Rendering and data splits**

We randomly sample 50 camera positions from a sphere with the object at the origin. For each model instance, 50 images are rendered using these camera positions with resolution of 224x224. We then split the data into

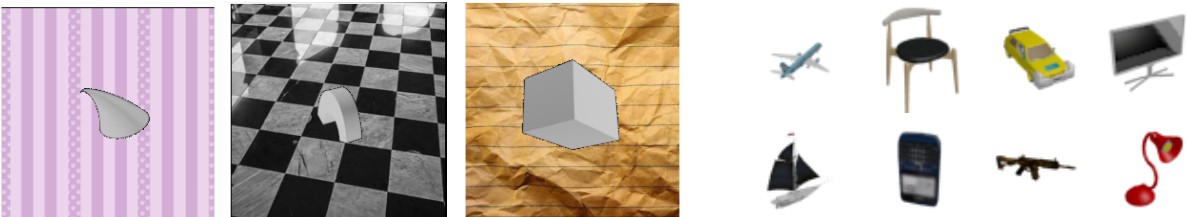

Figure 3: (Left) Example images from Geon3D with textured backgrounds. (Right) Example images from ShapeNet.

train/validation/test with ratio 8:1:1 using model instance ids, where each instance id corresponds to the scaling parameters described above. For more details of data preparation, see the Appendix.

## 4 General methods for experiments

**Pre-training** We use DVR and pixelNerf as our 3D reconstruction models. During 3D reconstruction pre-training, we first sample object instance ids of batch size, and then randomly sample a single view for each object instance to form a mini-batch, following the community convention of 3D reconstruction training. For the image encoder of 3D reconstruction models, we use ResNet18, which is expected to encode shape and category information during training. In the following Geon3D and ShapeNet experiments, we focus on the pre-training method that performs better 3D reconstruction on the respective dataset (e.g. DVR for Geon3D and pixelNeRF for ShapeNet.)

**Adversarial training** We used the python package [1] to perform adversarial training (AT) Madry et al. (2018). Throughout the experiments in this paper, we study a threat model where the adversary is constrained to $L_p$-bounded perturbations, where we use $p = \infty$ and $p = 2$. We consider the white-box setting, where we assume the adversary has complete knowledge of the model and its parameters. For AT-$L_2$ training, we train our models via Projected Gradient Decent (PGD) Madry et al. (2018) for 60 epochs with the batch size of 50, the attack steps of 7, the perturbation budget $\epsilon$ of 1.0, and the attack learning rate of 0.2. For AT-$L_\infty$ training, we train our models for 60 epochs with the batch size of 100, the attack steps of 7, the perturbation budget of 0.05, and the attack learning rate of 0.01. We use the best PGD step as an adversarial example during training. We use ResNet-18 (He et al., 2016) as our architecture throughout our experiments.

**Evaluation** It is notoriously difficult to correctly evaluate adversarial robustness Athalye et al. (2018). The attack based on Projected Gradient Descent (PGD) (Madry et al. (2018)) is widely used, but many defense methods are later found to be broken partly because PGD requires careful parameter tuning to be a reliable attack. To mitigate these issues, Croce & Hein (2020b) proposes AutoAttack, which is an ensemble of four strong, diverse attacks: two extensions of PGD, the white-box fast adaptive boundary (FAB) attack Croce & Hein (2020a), and the black-box Square Attack Andriushchenko et al. (2020). We use AutoAttack with the default parameter setting for both $L_\infty$ and $L_2$ robustness evaluation throughout our experiments.

## 5 Experiments using Geon3D

In this section, we will use the Geon3D shapes to create three increasingly more challenging datasets: *(i)* Geon3D with clean background ("Black Background"), *(ii)* Geon3D with randomly assigned textured backgrounds ("Textured Background"), *(iii)* Geon3D with correlated textured backgrounds, which introduces spurious correlations between background textures and categories ("Spurious Correlations"). For simplicity, we focus on 10 representative Geon categories (instead of the full 40 categories) and call it the Geon3D dataset. The dataset for adversarial training is a subset of the Geon3D data we used for 3D reconstruction pre-training.

---

[1]https://github.com/MadryLab/robustness

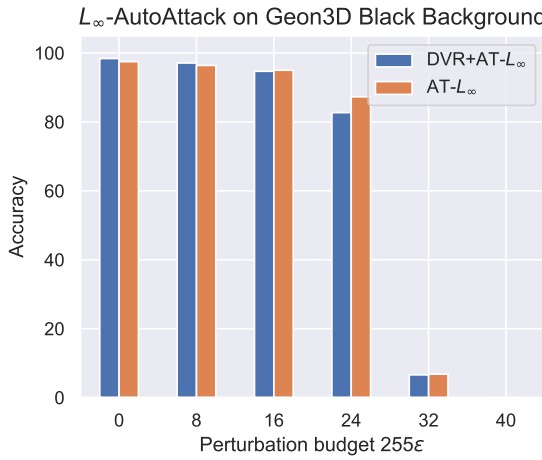 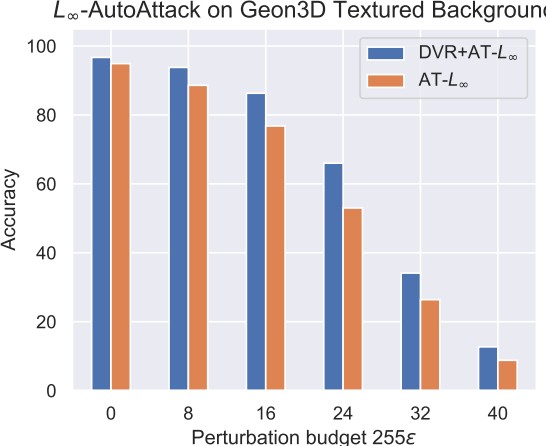

Figure 4: Adversarial robustness of vanilla adversarial training (AT) and 3D-based pre-training with increasing perturbation budget for $L_\infty$ threat model on Geon3D with black and textured backgrounds. DVR stands for Differentiable Volume Rendering. Between the simplest setting of Geon3D with black background and Geon3D with textured background, we observe that the effect of 3D reconstruction pre-training (DVR) emerges only under the latter, more complex data setting.

Specifically, we sample 10,000/1,000/1,000 images for train, validation, and test sets, respectively. We ensure that we sub-sample each split from the original train/val/test splits of Geon3D so that there is no data leakage from pre-training to adversarial training.

## 5.1 Adversarial robustness

**Setup** We start from the simplest setting: Geon3D with black background. We then vary the complexity of the experimental setting by introducing background textures to the dataset. Specifically, we replace each black background of Geon3D with a random texture image out of 10 texture categories chosen from the Describable Textures Dataset (DTD) (Cimpoi et al., 2014). Example images from this Geon3D Textured Background dataset can be seen in Figure 3 (Left). These two datasets allow us to analyze the effect of 3D reconstruction pre-training as a function of dataset (in particular, background) complexity.

**Results** Seen in Figure 4 are the results of adversarial robustness evaluation for $L_\infty$ threat models. For black background, DVR+AT slightly outperforms AT for $\epsilon = 8/255$ but as the the perturbation budget becomes large, AT outperforms DVR+AT. However, for textured background, DVR+AT consistently outperforms vanilla AT across all perturbation budgets. Figure 5 shows the results of adversarial robustness with $L_2$ threat models. On both the black and textured background settings, we find that AT is on the average, across all perturbation budgets, more robust than DVR+AT. However, consistent with the $L_\infty$ results, we see that DVR+AT better performs on the more complex textured background setting, slightly outperforming AT for small perturbation budgets.

## 5.2 Robustness to spurious correlations between shape and background

**Setup** As suggested by Ilyas et al. (2019), we can consider any image as a combination of robust and non-robust features. Robust features correspond to those that remain predictive even after corruption, and non-robust features are those that can flip model's decision under a certain perturbation budget. An example of such non-robust features for object recognition is background information, which can create spurious correlation that hurts model generalization Xiao et al. (2020). Adversarial training is one way to bias models to ignore non-robust features Ilyas et al. (2019). Here we test whether 3D-based pre-training, which directly targets robust features (scene geometry that causes pixel intensity values only on the foreground object), improves over vanilla AT, by more consistently ignoring non-robust features.

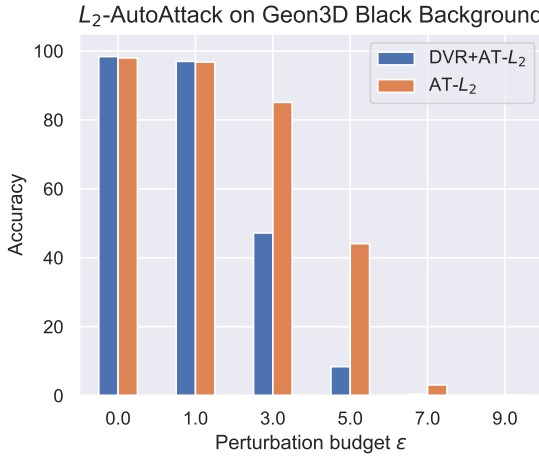 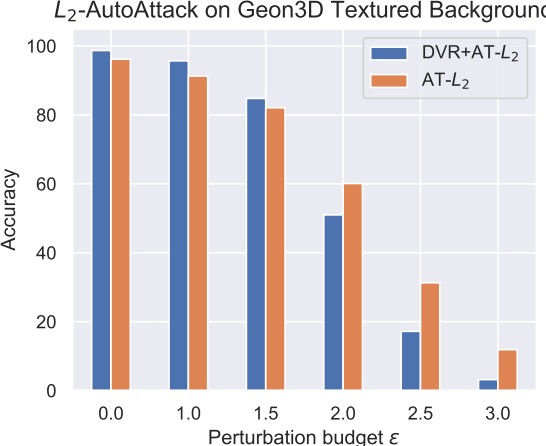

Figure 5: Adversarial robustness of AT and DVR+AT with increasing perturbation budget for $L_2$ threat models on Geon3D. Unlike the $L_\infty$ case, in the aggregate, 3D pre-training does not improve, and in fact lowers, the performance of AT. However, similar to the $L_\infty$ case, we continue to see the trend that 3D-based pre-training helps more as we increase the complexity of the data from black background to textured background.

To do this, we create a new variant of Geon3D, where we choose 10 texture categories from DTD and introduce spurious correlations between Geon category and textured background class (i.e., each Geon category is paired with one texture class). During 3D pre-training, we feed this dataset (referred to as "Correlated Texture") to the image encoder of the 3D reconstruction model. Adversarial training of all models are also performed using this dataset. Therefore, during adversarial training, a model can pick up classification signal from both the shape of Geon as well as background texture. To evaluate whether or not 3D pre-training helps models ignore spurious correlations more effectively, we prepare a test set that breaks the correlation between Geon category and background texture class by cyclically shifting the texture class from $i$ to $i + 1$ for $i = 0, ..., 9$, where the class 10 is mapped to the class 0. This design is inspired by (Geirhos et al., 2019), however, in our case, distributional shift from training to test set is designed to isolate out and directly measure the effect of 3D prior by fully disentangling the contributions of texture and shape.

**Results**   We note that in this section, we do not perform adversarial attacks, but simply evaluate all models on the newly constructed test set that breaks the correlation between textures and shape, as described above. The results, shown in Table 3, reveal two important insights.

First, we see a stark contrast between the two commonly used perturbations in adversarial training ($L_2$ vs. $L_\infty$). Our results suggest that $L_2$-based adversarial training biases model prediction to texture features, while $L_\infty$-based adversarial training biases model prediction to shape features. (Note that there is no inherent reason for a model to prefer texture to shape features, since the purpose of the model is to maximize its classification accuracy.) It seems such an observation is rarely discussed in the adversarial robustness literature, which we believe is a subject worthy of study in its own right in the context of aligning features used by machine learning and humans.

Second, we find that regardless of the perturbation set, DVR+AT outperforms AT, in fact by a large margin in the case of $L_2$ and still substantially for $L_\infty$. Together, these results suggest that we can view 3D-based pre-training as a way to bias models to prefer shape features, even in the presence of strong, spurious correlations.

**Summary: Improved robustness from 3D pre-training with increasing dataset complexity.**   We have varied the background texture and texture-shape correlation of Geon3D and measured how such variation affects the relationship between 3D-based pre-training and adversarial robustness. Our results with Geon3D so far seem to suggest that the benefit of 3D-based pre-training emerges as we move away from the simple

Table 3: Accuracy of adversarially-trained models against distributional shift in backgrounds. Here, all models are trained on Geon3D Correlated Textured (with background textures correlated with shape categories) and evaluated on a test set where we break this correlation. We see that for both $L_\infty$ and $L_2$, pre-training using DVR biases the models to prefer shape features to textures. Moreover, the difference between two threat models of vanilla AT suggests that AT-$L_2$ prefers texture features, while AT-$L_\infty$ prefers shape features.

| AT-$L_2$ | DVR+AT-$L_2$ | AT-$L_\infty$ | DVR+AT-$L_\infty$ |
|---|---|---|---|
| 10.8 | 35.6 | 79.0 | **84.20** |

setting (Geon3D Black Background); these patterns are summarized in Figure 6. Does this trend continue to hold even when we increase the complexity of object types? Next section answers this question using the ShapeNet dataset.

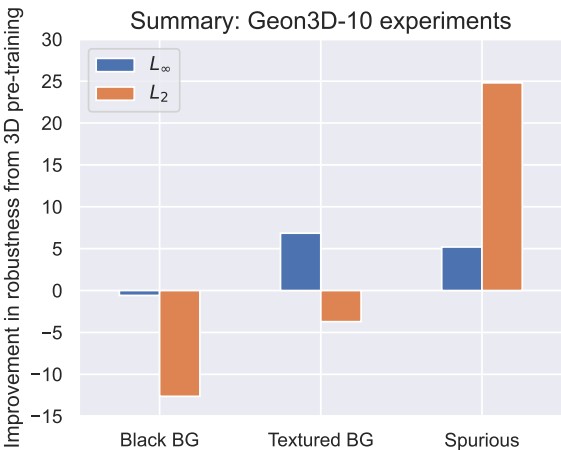

Figure 6: Average difference in robustness between DVR+AT and AT. With Geon3D, we see that the benefit of 3D pre-training emerges with increasing complexity of the dataset. We test and confirm this trend using ShapeNet in Section 6.

# 6 Experiments using more complex objects: ShapeNet

**Setup**  We use ShapeNet Chang et al. (2015) to evaluate the effect of 3D reconstruction pre-training on adversarial robustness under a shape distribution that is significantly more complex than Geon3D. Example images from ShapeNet are shown in Figure 3. We use the 13 most densely sampled shape categories from ShapeNet, as is commonly done in 3D reconstruction benchmarks. We perform 3D-based pre-training using the pixelNerf (PxN) model, which performs the basic task of 3D reconstruction more accurately than the DVR model on the ShapeNet dataset Yu et al. (2021). However, we note that we find similar results using DVR as the pre-training architecture (see Appendix). After 3D-based pre-training, we sub-sample 130,000/13,000/13,000 images as train/validation/test splits for adversarial training. We also ensure that object instances that are used for 3D reconstruction do not overlap with validation and test splits for adversarial training, so that there is no data leakage from pre-training.

**Results**  Figure 7 shows the results of adversarial robustness on ShapeNet. In contrast to previous results, we can see that for both $L_\infty$ and $L_2$ threat models, 3D-based pre-training (PxN+AT) improves over vanilla AT, across the entire range of perturbation budgets. This suggests that as we increase the complexity of object shapes, the 3D-based pre-training more consistently yields better robustness.

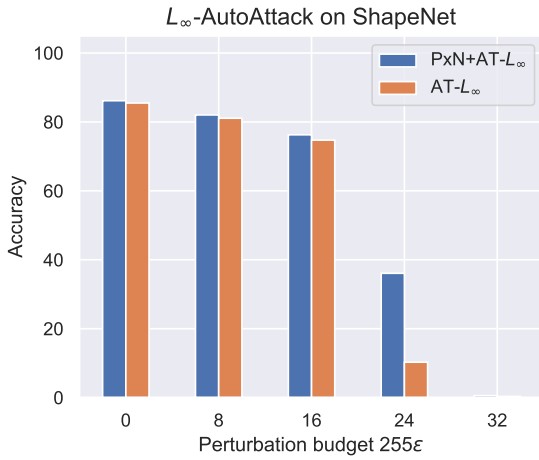 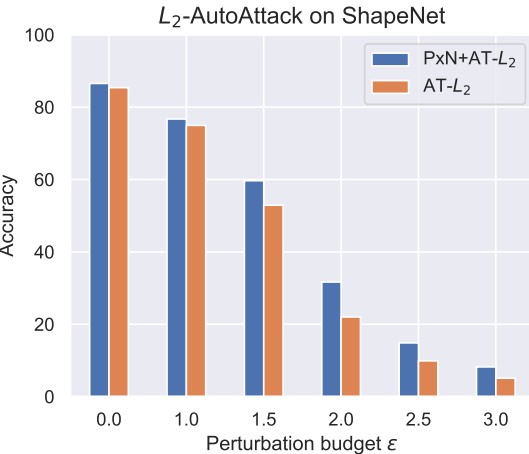

Figure 7: Adversarial robustness of AT and PxN+AT with increasing perturbation budget for ShapeNet. PxN stands for pixelNeRF. We see that 3D reconstruction pre-training (PxN+AT) improves over vanilla adversarial training (AT) for both $L_\infty$ and $L_2$ across all perturbation budgets.

## 7 3D-based vs. 2D-based pre-training

**Setup**   Does the improved adversarial robustness in the above section come from the fact that pre-training simply sees more data, or does the 3D inference needed during that pre-training for improved robustness? To answer this question, we compare 3D-based pre-training with 2D-based counterparts. Specifically, we use a simple AutoEncoder (AE) as well as a variational auto-encorder (VAE) Kingma & Welling (2014), and pre-train these models for an equivalent number of epochs as the 3D-based pre-training. We use ResNet-18 as an encoder for both AE and VAE (as we did for 3D-based pre-training). For adversarial training, we initialize the weights of the network from these ResNet-18 that are pre-trained via AE or VAE.

**Results**   We pick the perturbation budget $\epsilon = 24/255$ for $L_\infty$ and $\epsilon = 1.5$ for $L_2$, since those ranges show most difference between PxN+AT vs. vanilla AT in Figure 7. Table 4 shows the comparison between 2D and 3D-based pre-training. (See Appendix for a comparison across all perturbation budgets.) We see that pre-training based on AE and VAE does not necessarily improve over vanilla adversarial training, even though they produce high-quality reconstruction as shown in the appendix. And they both underperfom the 3D-based pre-training methods. These results show that the task of 3D reconstruction can encode more robust representations in the pre-trained weights than 2D-based pre-training.

Table 4: Adversarial accuracy comparison between 3D-based vs. 2D-based pre-training. We can see that 3D-based pre-training performs better than 2D-based counterparts in both $L_\infty$ and $L_2$ cases. Moreover, 2D-based pre-training is not necessarily helpful over the vanilla AT as can be seen for the $L_2$ case. AE stands for 2D-based Auto Encoder, VAE stands for Variational AutoEncoder.

|  | AT | AE+AT | VAE+AT | DVR+AT | PxN+AT |
|---|---|---|---|---|---|
| $L_\infty(\epsilon = 24/255)$ | 10.28 | 25.56 | 8.71 | 28.54 | **36.08** |
| $L_2(\epsilon = 1.5)$ | 52.9 | 47.95 | 45.04 | 54.11 | **59.67** |

## 8 Limitations

In this paper, we view 3D reconstruction as a pre-training task that provides better weight initialization in the form of a 3D object prior. The robustness gained from such a 3D prior is necessarily constrained by the capability of the underlying 3D reconstruction models. We studied only one form of causal, thus by definition robust set of features (3D shape and pose); future work should consider incorporating priors based on other

causal variables, such as the physical properties of objects. We studied only one way to induce such a prior (via pre-training); future work should explore other ways in which explicit robust properties can be integrated to AT. Finally, future work should understand why 3D pre-training is not helpful for the simplest data setting studied here.

## 9 Related Work

**Pre-training for adversarial training.** Hendrycks et al. (2019) proposes pre-training to improve adversarial robustness but their work focuses on classification-based pre-training by introducing more natural images. In contrast, our work uses pre-training to encode a prior about 3D object shape and pose.

**Shape bias to induce robustness**. A recent line of work explores methods to increase *shape bias* as a way to make neural network models more robust to image perturbations Geirhos et al. (2019); Wang et al. (2018; 2019). A notable example is given by Geirhos et al. (2019), who proposes to train a model on Stylized-ImageNet (SIN), which are created by imposing various painting styles to images from ImageNet Deng et al. (2009). Unlike these studies, that indirectly tackle shape-bias by reducing the reliance on texture, our work induces shape bias directly into image classifiers, via 3D reconstruction pre-training.

**3D datasets**. Geon3D is smaller in scale and less complex in shape variation relative to some of the existing 3D model datasets, including ShapeNet (Chang et al., 2015) and ModelNet (Zhirong Wu et al., 2015). These datasets have been instrumental for recent advances in 3D computer vision models (e.g. Niemeyer et al. (2020); Sitzmann et al. (2019)). As we demonstrate in this work, Geon3D allows us to systematically study the relationship between 3D-based pre-training and adversarial training by varying the complexity of the dataset, bridging toy datasets to more realistic datasets such as ShapeNet.

## 10 Conclusions

We investigated how 3D-based pre-training can affect robust accuracy of adversarial training. We start from the simplest setting: Geon3D with black background. In this case 3D-based pre-training does not improve vanilla adversarial training. However, we find that 3D-based pre-training improves over adversarial training under more complex data distributions, including the ShapeNet objects. Importantly, 3D-based pre-training outperforms 2D-based pre-training methods that otherwise receive identical training procedures. We also identify that $L_2$ adversarial training tends to prefer texture features while $L_\infty$ adversarial training biases the model to use shape features. We hope that these results motivate further exploration of 3D vision for addressing adversarial robustness.

### Broader Impact Statement

While not intended to cause any harm, improved shape perception system could be misused to increase worker surveillance, causing detrimental effect on people's autonomy and privacy at work. Furthermore, when combined with face recognition algorithm, improved shape bias of vision models could further advance unethical use such as distinguishing faces of a certain ethic group from those of other ethnicity.

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

## A    Dataset

A line of work in psychophysics of human visual cognition have argued that the visual system exploits certain types of shape features in inferring 3D structure and geometry. In Geon3D, by treating these shape features as the dimensions of variation, we model 40 classes of 3D objects, and render them from random viewpoints, resulting in an image set and their corresponding camera matrices. Upon publication, we will make the Geon3D dataset publicly available. Until then, we hope Figure 8 suffices to get a overview of the dataset.

**List of 40 Geons**   In Figure 8, we provide a list of 40 Geons we have constructed. The label for each Geon class represents the four defining shape features, in the order of "axis", "cross section", "sweep function", "termination", as described in the main paper. We put "na" for the termination when the sweep function is constant. We also distinguish the two termination types "c-inc" and "c-dec" when the sweep function is monotonic. For instance, "c-inc" means that the curved surface is at the end of the increasing sweep function, whereas "c-dec" means that the curved surface is at the end of the decreasing sweep function. As a reference, here is the mapping between the name and the code of 10 Geons we used in 10-Geon classification: "Arch": `c_s_c_na`, "Barrel": `s_c_ec_t`, "Cone": `s_c_m_p`, "Cuboid": `s_s_c_na`, "Cylinder": `s_c_c_na`, "Truncated cone": `s_c_m_t`, "Handle": `c_c_c_na`, "Expanded Handle": `c_c_m_t`, "Horn": `c_c_m_p`, "Truncated pyramid": `s_s_m_t`.

## B    Additional Results

### B.1    Adversarial Robustness on ShapeNet

In Figure 9, we show additional results of adversarial robustness for both $L_infty$ and $L_2$ threat models. In addition to PxN+AT, we include DVR+AT. We also include AE+AT and VAE+AT across the perturbations we tested. We see that 3D-based pre-training (PxN+AT and DVR+AT) outperforms 2D-based pre-training (AE+AT and VAE+AT) as we increase the magnitude of the perturbations $\epsilon$. We note that for $L_\infty$, VAE+AT is the best in terms of the clean accuracy ($\epsilon = 0$), which makes VAE+AT also the best-performing model for $\epsilon = 8/255$ but for larger perturbations, DVR+AT and PxN+AT outperform AT+VAE. Similarly, for $L_2$, AE is the best in terms of the clean accuracy ($\epsilon = 0$) but again for larger perturbations, DVRA+AT and PxN+AT outperform other models.

Seen in Figure 10 are the reconstructed images of AutoEncoder and Variational AutoEncoder (VAE).

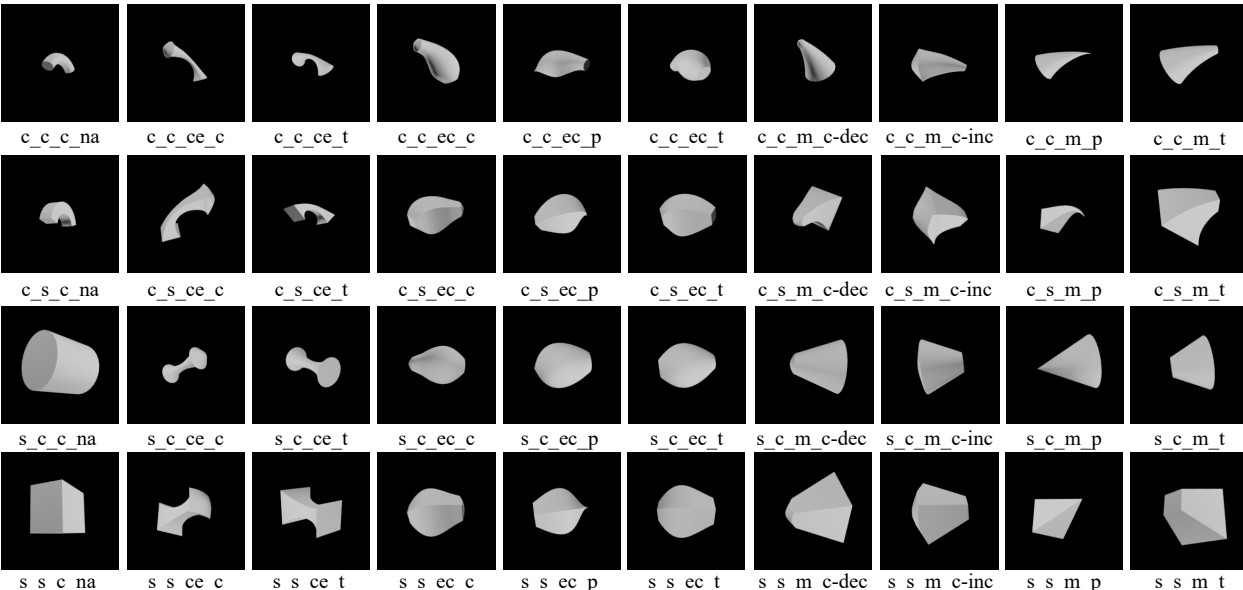

Figure 8: The list of 40 Geons we constructed.

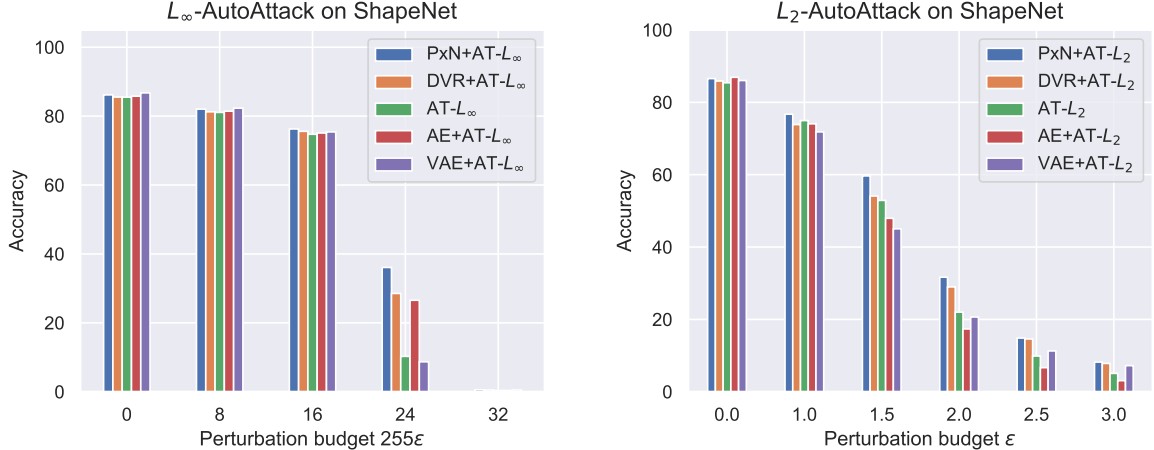

Figure 9: Adversarial robustness comparison between PxN+AT, DVR+AT, AE+AT, VAE+AT, and AT for both $L_\infty$ and $L_2$ threat models with increasing perturbation budget $\epsilon$ on ShapeNet.

## C  Details of 3D reconstruction training

We provide details of the problem setup of 3D reconstruction, following (Niemeyer et al., 2020).

During training, we render an image, which is then used to minimize the RGB reconstruction loss. To render a pixel of an image observed by a virtual camera, we need to first find the world coordinate of the intersection of the camera ray with the object surface, and then map the world coordinate into a RGB color.

Let $u = (u_1, u_2)$ be the image coordinate of the pixel we want to render. To find the world coordinates of the intersection, we first parameterize the points along the camera ray $r_{p_0 \to (u_1, u_2)}$ by the distance $d$ to the camera origin $p_0$ as follows:

AutoEncoder Reconstruction                                    VAE Reconstruction

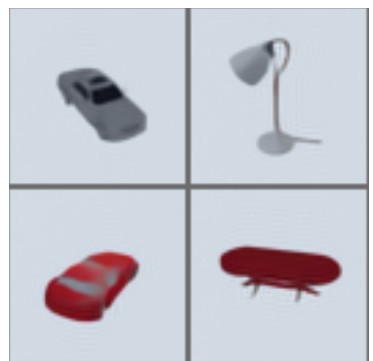    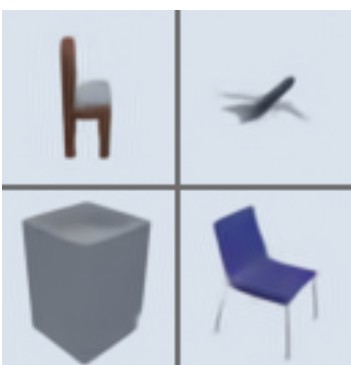

Figure 10: Reconstructed ShapeNet images. (Left) AutoEncoder, (Right) VAE.

$$r_{p_0 \to (u_1, u_2)}(d) = R^T \left( K^{-1} \begin{pmatrix} u_1 \\ u_2 \\ d \end{pmatrix} - T \right)$$

Here, $R \in \mathbb{R}^{3 \times 3}$ is a camera rotation matrix, $T \in \mathbb{R}^3$ is a translation vector, and $K \in \mathbb{R}^{3 \times 3}$ is a camera intrinsic matrix. In the main paper, we denote $c^{ex} = [R, T]$, and $c^{in} = K$. Here, $T$ is the position of the origin of the world coordinate system with respect to the camera coordinate system. Therefore, the position of the camera origin $p_0$ (w.r.t. the world coordinate system) is $-R^T T$.

Then we solve the following optimization problem:

$$\text{argmin} \quad d \quad \text{s.t.} \quad r_{p_0 \to (u_1, u_2)}(d) \in \Omega \tag{1}$$

where $\Omega$ is the set of points $p$ in $\mathbb{R}^3$ such that $f_\theta(p) = 0.5$.

To solve for $d$, we start from the camera origin $p_0$ and step along the ray until object surface is intersected, which we can determine by evaluating the points along the ray via $f_\theta$.

To summarize, we are given a set of object images $\{x_i \in \mathbb{R}^{H \times W \times 3}\}_{i=1}^n$, their corresponding binary object masks $\{m_i \in \mathbb{R}^{H \times W}\}_{i=1}^n$, and extrinsic/intrinsic camera matrices $\{c_i = (c_i^{ex} \in \mathbb{R}^{3 \times 3} \times \mathbb{R}^3, c_i^{in} \in \mathbb{R}^{3 \times 3})\}_{i=1}^n$. Let $\mathcal{U}_0$ be a set of pixel points which lie inside the ground truth object mask and where the model predicts a depth. $\mathcal{U}_1$ is a set of points outside the object mask where the model falsely predicts depth. Finally $\mathcal{U}_2$ is a set of points inside the object mask where the model does not predict any depth. Then the objective is:

$$\underset{\phi, \theta, \theta'}{\arg \min} \, \mathbb{E} \Big[ \sum_{u \in \mathcal{U}_0} (||\hat{x}_u - x_u||_1 + \lambda_1 \mathcal{L}_{\text{normal}}(\hat{p}_{u,c}|z))$$

$$+ \lambda_2 \sum_{u \in \mathcal{U}_1} \text{BCE}(f_\theta(\hat{p}_{u,c}|z), 0) + \lambda_3 \sum_{u \in \mathcal{U}_2} \text{BCE}(f_\theta(p_{\text{rand}(u),c}|z), 1) \Big]$$

Here, BCE stands for Binary Cross Entropy loss, and $\hat{p}_{u,c} = r_{p_0 \to u}(\hat{d})$, where $\hat{d}$ is the predicted depth, provided as a solution to the optimization problem 1. The value of $p_{\text{rand}(u),c} = r_{p_0 \to u}(d_{\text{rand}(u)})$, where the value of $d_{\text{rand}}(u)$ is chosen uniformly randomly on the ray to encourage occupancy for $u \in \mathcal{U}_2$. $\hat{x}_u = r_{\theta'}(\hat{p}_{u,c}|z)$ for $u \in \mathcal{U}_0$. $z = g_\phi(x_i^{(rand)})$, where we take a random view $x_i^{(rand)}$ from the same object instance as $x_i$.

$\mathcal{L}_{\text{normal}}(p|z)$ is the normal loss, which is a geometric regularizer to encourage smooth object surface. For a point $p \in \mathbb{R}^3$ and some object encoding $z$, the unit normal vector can be calculated by:

$$n_\theta(p|z) = \frac{\nabla_p f_\theta(p|z)}{||\nabla_p f_\theta(p|z)||_2}$$

We apply the $l_2$ loss to minimize the difference between the normal vectors at $p$ and $p'$, where $p'$ is in a small neighbourhood around $p$. Formally,

$$\mathcal{L}_{\text{normal}}(p|z) = ||n_\theta(p|z) - n_\theta(p'|z)||_2$$

for a point $p \in \mathbb{R}^3$.

## D    Additional Training details

We used Tesla V100 GPUs for all of our experiments. DVR 3D reconstruction training takes roughly about 1.5 days on a single GPU. The hyperparameters for adversarial training, described in the main paper, were chosen by monitoring the model convergence on the validation set. All the other results are from a single training run and a single evaluation run.

**DVR**  We used the code [2] open-sourced by Niemeyer et al. (2020). We followed the default hyperparameters recommended by Niemeyer et al. (2020) for 3D reconstruction training, with the exception of batch size, which we set 32 to fit into a single GPU memory.

**PixelNeRF**  We use the code [3] open-sourced by the original authors Yu et al. (2021).

**AE and VAE**  We use the code [4] from pytorch-lightning bolts to train AE and VAE on ShapeNet. Both encoder and decoder are based on ResNet18.

**Dataset**  For training Geon3D image classifiers, we center and re-scale the color values of Geon3D with $\mu = [0.485, 0.456, 0.406]$ and $\sigma = [0.229, 0.224, 0.225]$, which is estimated from ImageNet. We construct the 40 3D model instances as well as the whole training data in Blender. We then normalize the object bounding box to a unit cube, which is represented as `1.0_1.0_1.0` in the dataset folder.

**Background textures**  We used the following label-to-texture class mapping: {0: 'zigzagged', 1: 'banded', 2: 'wrinkled', 3: 'striped', 4: 'grid', 5: 'polka-dotted', 6: 'chequered', 7: 'blotchy', 8: 'lacelike', 9: 'crystalline' }. For the distributional shift experiment we used the following mapping: { 0: 'crystalline', 1: 'zigzagged', 2: 'banded', 3: 'wrinkled', 4: 'striped', 5: 'grid', 6: 'polka-dotted', 7: 'chequered', 8: 'blotchy', 9: 'lacelike', }. The DTD data is licensed under the Creative Commons Attribution 4.0 License. [5]

---

[2]`https://github.com/autonomousvision/differentiable_volumetric_rendering`
[3]`https://github.com/sxyu/pixel-nerf`
[4]`https://pytorch-lightning-bolts.readthedocs.io/en/latest/autoencoders.html`
[5]https://creativecommons.org/licenses/by/4.0/, https://www.tensorflow.org/datasets/catalog/dtd

