# OpenReview forum: "On 3D Reconstruction Pre-training to Improve Adversarial Robustness"
_TMLR — Rejected by TMLR_

### Review · Reviewer_nSUm · 2023-01-09

**Summary Of Contributions:**

The authors introduce a new dataset where 3D objects are rendered from random viewpoints and contain shape variations. Through taking 3D-based reconstruction on the new dataset as the pre-training task, the adversarial robustness of networks on classification task can be improved. The authors conduct experiments on different datasets and include comparisons of different pre-training tasks to demonstrate the effectiveness of 3D-based pre-training.

**Audience:**

Yes

**Broader Impact Concerns:**

No concerns.

**Claims And Evidence:**

Yes

**Requested Changes:**

1. The authors are recommended to demonstrate the advantage of introduced Geon3D. For example, taking 3D-based reconstruction task, how is Geon3D compared with other datasets in terms of the adversarial accuracy or pre-training efficiency?

2. More comparisons with other existing robust pre-training techniques can make the results more promising.

3. The authors should discussion other recent defense techniques in their related work.


**Strengths And Weaknesses:**

Pros:

++ Introducing a new dataset for 3D-based reconstruction pre-training task seems interesting and could benefit the adversarial robustness community.

++ The evaluation is conducted under various settings, such as different backgrounds, datasets with different complexities, and different perturbation budgets.

Cons:

-- Although the authors introduce a new dataset, it is difficult to see the advantage of proposed Geon3D over other existing 3D model datasets in pretraining phase.

-- Utilizing pre-training as a defense technique is not a new story and the comparisons with other robust pre-training techniques are missing [a, b].

-- Besides pre-training for weight initialization, the authors should include more discussion of other recent defense techniques [c, d].

[a]. Using Pre-Training Can Improve Model Robustness and Uncertainty. ICML 2019.

[b]. Robust Pre-Training by Adversarial Contrastive Learning. NeurIPS 2020.

[c]. Robustness in deep learning: The good (width), the bad (depth), and the ugly (initialization). NeurIPS 2022.

[d]. Random Normalization Aggregation for Adversarial Defense. NeurIPS 2022.

---

### Review · Reviewer_t13W · 2023-01-10

**Summary Of Contributions:**

This work studies how pre-training with 3D reconstruction can improve adversarial robustness of 2D image classification models. Adversarial training is known as an effective technique to mitigate the threat of adversarial attacks. This work builds upon adversarial training and introduces 3D reconstruction pre-training to learn robust image features to overcome adversarial attacks. To evaluate the proposed approach, a new dataset Geon3D is constructed with simple shapes with the concept of geons (geometric ions). This work discovers that, as the complexity of the dataset increases, 3D-based pre-training serves as an effective approach to improve adversarial robustness over simple adversarial training.

**Audience:**

Yes

**Broader Impact Concerns:**

No concerns.

**Claims And Evidence:**

Yes

**Requested Changes:**

In addition to the weaknesses (see details above), these minor issues might be fixed:

- Section 2.1: Citation for NeRF is missing.

- Figure 4: On Geon3D Black Background, there is a significant accuracy drop (for both AT and DVR+AT) when increasing the perturbation budget from 24 to 32, leading to almost zero accuracy. However, on Geon3D Textured Background (which is assumed to be a more complex dataset), there is no such a significant drop, and the accuracy is still decent after the perturbation is increased to 32. Is there any intuitive explanation for the accuracy drop?

- Appendix B.1: There is an incorrect notation “$L_{infty}$” in the first line.


**Strengths And Weaknesses:**

Strengths:

- Various dataset settings are evaluated: a) simple shapes without background, 2) simple shapes with textured background, 3) spurious correlations between shape and background texture, and 4) complex shapes. This work shows that as the dataset gets more and more complex, 3D-based pre-training can more effectively improve adversarial robustness.

- This work proposes a novel dataset Geon3D consisting of images rendered from simple 3D shapes (geons). This dataset might be helpful for future research.

- This work shows an interesting finding that “$L_2$-based adversarial training biases model prediction to texture features, while $L_\infty$-based adversarial training biases model prediction to shape features.” This finding might be inspiring to the research of adversarial robustness.

Weaknesses:

- This work has mainly compared “adversarial training”and “3D-based pre-training + adversarial training.” To better demonstrate the benefit of 3D-based pre-training in robustness, it might be good to disentangle adversarial training and show comparison between “standard classification training” and “3D-based pre-training + standard classification training” as well.

- ResNet-18 is used as the base vision model throughout this work. It might be good to experiment with more backbones and show readers how 3D-based pre-training interacts with the complexity of vision models.

- The proposed approach involves an additional pre-training stage, which might introduce efficiency concerns. Another question is, if the baseline (adversarial training without pre-training) is allowed equal total training time (e.g., by extending the training schedule) as the proposed approach (3D-based pre-training + adversarial training), will the proposed approach still outperform it?

- Another practicality concern: A series of datasets with increasing complexity are evaluated in this work, and 3D-based pre-training may have negative effects when the dataset is simpler (see Figures 4&5). In practice, how to decide if the target dataset is complex enough for applying this 3D-based pre-training for improved adversarial robustness? Also, all the evaluated datasets are based on synthesized images of shapes. Will this approach be applicable to real-world images?

---

### Review · Reviewer_GdK2 · 2023-02-05

**Summary Of Contributions:**

1. The paper raises an interesting question: can 3D reconstruction pre-training improve adversarial robustness? The question is motivated by some reasonable analysis. For example, existing works have shown that adversarial training enhances robustness by learning robust features, and 3D reconstruction pre-training encodes causal information underlying the formation of images, therefore it should be able to help adversarial training learn more robust features and further improve the model’s robustness. Inspiration from human vision is also discussed.

2. To answer the research question, a novel dataset is proposed. The dataset is simple but plays an important role in the work for reaching some interesting results.

3. The authors conduct extensive experiments and conclude that under certain situations, 3D reconstruction pre-training can benefit the adversarial robustness of downstream 3D object classification models.

**Audience:**

Yes

**Broader Impact Concerns:**

I have no concerns about the ethical implications of the work.

**Claims And Evidence:**

Yes

**Requested Changes:**

1. [Critical] Adversarially train the baselines for a longer time, or pre-train the baseline using the evaluation task (without adversarial training) for some equivalent training time to the 3D reconstruction pretraining.

2. [Critical] Discuss the motivation of the evaluation task. Explain why it is used to evaluate adversarial robustness and why other tasks and datasets are not suitable. Also, describe more details of the finetuning, such as the loss function. It may also be helpful if the performance of models that are not trained adversarially is shown.

3. [Not critical] Modify the narratives in the introduction so that the readers can understand how the adversarial robustness is evaluated at the beginning.

4. [Not critical] Section D shows the pretraining time (1.5 days on one V100 GPU), but the pretraining epoch number is not shown.

5. [Minor] First Line of B.1 $L_infty$ -> $L_{\infty}$.

**Strengths And Weaknesses:**

Strengths：

1. The motivation of the paper is inspiring and exciting.

2. The paper proposed a novel dataset that is simple but useful. The design of the dataset is backed by classical concepts and theories in cognitive psychology.

3. The state-of-the-art adaptive attack method is used for evaluation, making the results more reliable.

4. The experimental results are consistent and have some potential for discovering important insights with further investigation.


Weaknesses:

1. The baseline is too weak.

The common practice in the pretraining literature is that the baseline is finetuned with a higher cost than the model with pretraining. For example, in [1], the baseline is finetuned for 200 epochs while the pretrained model is finetuned for 50 epochs. Also, the baseline in [2] is finetuned for 99 epochs while models with pretraining are finetuned for less than 50 epochs.

In contrast, in this work, it seems both the baseline and the model with pretraining is adversarially trained for 60 epochs. Therefore, it is unclear if the improvement is from the 3D reconstruction pretraining or simply more data and more training time. The experiment in Section 7 that compares the proposed pretraining with 2D-based pretraining using AE and VAE is not convincing because apparently not all pretraining tasks are effective for improving robustness and 2D-based generative pretraining might just be one of the ineffective ones.

2. The introductory part is somewhat misleading.

The narrative in Section 1 made me think that the “adversarial robustness” improved by 3D reconstruction pretraining is the adversarial robustness on 2D datasets like CIFAR-10 or ImageNet. For example, the term “robust/non-robust features” is from Ilyas et al. (2019) based on 2D images. Moreover, in the fifth paragraph, the author stated that standard benchmarks are not designed for 3D reconstruction tasks, hence using another two 3D datasets. However, since the finetuning/evaluation task is not 3D reconstruction, I expect the paper to use these standard benchmarks to evaluate the performance.

3. The evaluation task requires more discussion.

Besides Figure 1(b), I find no information on the finetuning/evaluation task, i.e., the classification of 2D rendered images of 3D objects. I am not sure it is an important task as I cannot find any paper that performs this task on ShapeNet or any other 3D datasets.

[1] He et al., Masked Autoencoders Are Scalable Vision Learners. 2021.

[2] Chen et al., Adversarial Robustness: From Self-Supervised Pre-Training to Fine-Tuning, 2020.

---

### Decision · Action_Editors · 2023-03-18

**Recommendation:** Reject

**Comment:**

This paper studies an interesting problem that whether a 3D reconstruction pre-training can improve adversarial robustness. But all reviewers still hold major concerns about this paper after the authors' response and were leaning reject.

In particular, the reviewers thought the authors only provided a plain discussion of the advantages over other 3D datasets, rather than a thorough experimental comparison of the performance on the authors' new dataset and those existing datasets. The reviewers requested an evaluation of adversarial robustness on standard 2D image datasets, which would facilitate an easy comparison with previous baselines. The authors' explanation of weaker generalisation as a cause of worse accuracy drop was not convincing enough without any experimental support. At last, the reviewers thought the authors need to refer to other popular pre-training and fine-tunning settings in the literature (e.g., in terms of epochs) for a much strong and fair comparison, and the introduction about the evaluation tasks shall be further clarified to avoid the potential misunderstanding. All these remaining major concerns are quite critical if the authors consider a future resubmission.

**Audience:**

The motivation of this paper is inspiring. But per reviewers' comments, the paper itself is not in a good enough status yet. The proposed new dataset is interesting, but if its advantages over existing 3D datasets are unclear, the audience could not have a strong motivation to try this unmatured dataset, compared with those well-known datasets in the literature. Regarding the 3D reconstruction pre-training algorithm, its performance in experiments could be questionable, e.g., the significant accuracy drop under a larger perturbation, the inappropriate setup of pre-training and fine-tuning training epochs, and the kind of misleading statement about the evaluation. These facts would hurt the audience's interest in the paper.

**Claims And Evidence:**

There are quite a few claims made in the submission that are not well supported with strong enough evidence. For example, reviewers criticised the significant accuracy drop when attacked by larger perturbations. The authors gave their own guess and thought the reason might be the weaker generalisation. But there is no relevant study to justify this claim, and the worse accuracy would directly challenge the practical use of the proposed algorithm.
Though in the authors' response, the potential advantage of the proposed dataset over other 3D datasets was discussed.  But nearly all the relevant discussions here were qualitative. It would be much more convincing if the authors can directly align the pre-training results on both their own dataset and other 3D datasets and present them in a quantitative table/figure.